# Experimental Study and Artificial Neural Network Simulation of Cutting Forces and Delamination Analysis in GFRP Drilling

**DOI:** 10.3390/ma15238597

**Published:** 2022-12-02

**Authors:** Katarzyna Biruk-Urban, Paul Bere, Jerzy Józwik, Michał Leleń

**Affiliations:** 1Department of Production Engineering, Mechanical Engineering Faculty, Lublin University of Technology, 20-618 Lublin, Poland; 2Department of Manufacturing Engineering, Faculty of Industrial Engineering, Robotics and Production Management, Technical University of Cluj-Napoca, Memorandumului 28, 400114 Cluj-Napoca, Romania

**Keywords:** GFRP, drilling, cutting force, delamination

## Abstract

This paper reports the results of measurements of cutting forces and delamination in drilling of Glass-Fiber-Reinforced Polymer (GFRP) composites. Four different types of GFRP composites were tested, made by a different manufacturing method and had a different fiber type, weight fraction (wf) ratio, number of layers, but the same stacking sequence. GFRP samples were made using two technologies: a novel method based on the use of a specially designed pressing device and hand lay-up and vacuum bag technology process. The study was conducted with variable technological parameters: cutting speed v_c_ and feed per tooth f_z_. The two-edge carbide diamond-coated drill produced by Seco Company was used in the experiments. Cutting-force components and delamination factor were measured in the experiments, and photos of the holes were taken to determine the delamination. In addition, modeling of cause-and-effect relationships between the technological drilling parameters v_c_ and f_z_ was simulated with the use of artificial neural network modeling. For all tested GFRP materials, an increase in f_z_ led to an increase in the amplitude of cutting-force component F_z_. The lowest values of the amplitude of cutting-force component F_z_ were obtained with the lowest tested feed per tooth value of 0.04 mm/tooth for all tested materials. It was observed that materials produced with the use of the specially designed pressing device were characterized by lower values of the cutting-force component F_z_. It was also found that the delamination factor increased with an increase in f_z_ for all tested GFRP materials. A comparison of the lowest and the highest values of f_z_ revealed that the lowest delamination factor increase was archived by the B1 material and amounted to about 12.5%. The error margin of the obtained numerical modeling results does not exceed 15%, so it can be concluded that artificial neural networks are a suitable tool for modeling cutting force amplitudes as a function of v_c_ and f_z_. The study has shown that the use of the special pressing device during the manufacturing of composite materials has a positive effect on delamination.

## 1. Introduction

GFRP is a composite material, whose structure is made of a minimum of two components: Glass Fiber (GF) and polymer (usually an epoxy) [1]. GFRP characteristics depend on fiber properties, their type and amount, layers arrangement, chemical stability, matrix strength and interface bonding [2]. Due to their unique properties, such as high mechanical properties, low density, high strength and corrosion resistance [3,4] these materials have been widely used in many industries since the beginning of the 20th century. The most widespread applications for GFRPs include aerospace, aviation, automotive, wind turbine blades, as well as furniture, etc. [5]. At the end of the manufacturing process of composite components, machining operations are often required to ensure required tolerances and to create fitting and joining surfaces [6] with the use of bolts or rivets [7]. There are many machining processes used in the production of GFRP parts, including conventional machining turning [8], milling [9] and drilling [10] as well as non-conventional machining processes such as laser machining [11], water jet cutting [12], ultrasonic machining, electrochemical machining, electrical discharge machining, chemical machining and photochemical machining [13].

The aviation industry is one of the industries in which drilling accounts for 40% of all machining operations in components assembly (riveting, screwing) [14]. Aircrafts can have over one hundred thousand mounting holes. High requirements are imposed on holes for rivet and bolt connections, as only defect-free, undamaged and precise holes can ensure strength and precision of the connections. Therefore, research should be conducted to investigate the behavior of GFRP composites during the machining processes, especially with regard to drilling.

Research works are carried out on various types of drilling, such as conventional drilling [15], grinding drilling [16], vibration-assisted drilling [17] and high-speed drilling [18]. There is no doubt that conventional drilling is the most popular and widely used method. Previous studies on conventional drilling have focused on the influence of technological drilling parameters, tool geometry [19] and material type on the cutting forces and workpiece failures such as fiber pull-out, fiber breakout, matrix smearing and delamination (which impacts about 60% of aircraft components [20]) in order to determine the optimum cutting parameters to prevent the occurrence of these failures.

Delamination affects the structural integrity and reliability of FRP composites; therefore, it has a significant economic impact, especially when considering different stages associated with components assembly [21]. In drilling FRP composite laminates, delamination was caused by a combination of mechanical and thermal damage. It occurred at the entry (peel-up delamination) and exit planes (push-out delamination) of the composite laminate. Peel-up delamination is caused by two fracture mechanisms. The first one occurs when the fibers of the upper layers are not cut properly due to inadequate cutting conditions. The other mechanism occurs when the cutting edges of the twist drill contact the laminate surface. A peeling force is induced through the slope of a drill flute and separates the top layers, causing peel-up delamination [22]. Push-out delamination of the bottom surface is caused by both failure mechanisms because the drilled composite is subjected to an axial force and bending [23].

A study on GFRP drilling carried out by K. Siva Prasad [22] demonstrated that delamination depended on factors such as feed rate, fiber orientation and spindle speed, among others. The minimum peel-up delamination was obtained with the following machining parameters: a speed of 400 rpm, a feed rate of 0.02 mm/rev, sample thickness of 8 mm and a fiber orientation of 0°. T. Panneerselvam et al. [24,25] analyzed the drilling parameters of a Sisal GFRP and their influence on the delamination. On the basis of the conducted research, it was found the parameter that has the greatest impact on peel-up delamination is the feed rate. Drilling with higher spindle speeds and lower feeds resulted in minimum delamination due to a reduction in thrust force and torque in GFRP drilling. The machinability of GFRP pipes was also studied by L. Gemi in [25]. In the study, different tools were used to assess their impact on cutting forces and delamination. The brad and spur drill and brad-center drill used in research decreased cutting forces by 8% and 13% on average, in comparison with a conventional twist drill. It was also found that the brad-center drill caused less delamination. The study of R. Bhat et al. [26] investigated the influence of technological parameters such as feed rate, speed and GFRP composite thickness on peel-up and push-down delamination and surface roughness. The results showed a significant impact of sample thickness on the overall performance index defining the machining quality of a drilled hole, contributing 21.30% to the variance, while feed was the second determining factor. B. Singaravel et al. [27] studied the influence of different types of drill bits and drilling parameters, including feed rate and spindle speed, on the hole-quality parameters of a drilled hole. One of the parameters used to assess hole quality was delamination. It was found that the factors influencing delamination were spur drill, higher speed and lower feed rate. Another study [28] also investigated the effect of different drilling parameters (cutting speeds and feeds) on the delamination of GFRP materials. A comparison was made between the results of the experiment and the results of the finite element model tests. The FEM models confirmed the experimental results. The optimum technological drilling parameters defined for the tool were a speed of 700 rpm and a feed rate of 0.4 mm/rev. E. Kilickap [29] found that delamination is influenced by higher cutting speeds and feed rates in dry drilling of GFRP composites. U. A. Khashaba [23] in the paper analyzed the influence of machining parameters on heat zone, thrust force, torque, and delamination in drilling GFRP composite laminates. It was found that the delamination is proportionally dependent on the thrust force and inversely dependent on the temperature.

The aim and novelty of the study presented in this paper was to assess the influence of the different technological drilling parameters on the cutting forces occurring in this process and the delamination of GFRP composites plates. The GFRP plates were made of two different manufacturing technologies: a traditional one, vacuum bag method and a new innovative one which was patented [30]. The results of the obtained parameters in the drilling process of the GFRP plates obtained by two different technologies were compared. The machinability of those materials was not the subject of any research. The knowledge of the machinability of these new materials is crucial from the point of view of their industrial applications because the selection technological parameters of drilling affects cutting forces, and thus the quality of drilled holes.

## 2. Materials and Methods

The purpose of this study was to investigate the effect of selected technological drilling parameters (cutting speed v_c_ and feed per tooth f_z_) and different GFRP materials on the cutting forces and their influence on delamination. On this basis, it will be possible to define the optimum technological parameters for drilling GFRPs manufactured with the new method. The applied research plan is presented in Figure 1.

### 2.1. Materials

Four different types of GFRP plates were manufactured and investigated in this study. For plates noted by “A” GF Twill 2 × 2 woven type by 280 g/m^2^, EC9-3×68 Tex yarn type in warp and two different number of layers were used. For plates noted by “B”, GF type plain woven by 300 g/m^2^ was used. The warp of the woven used was rowing’s by 300/300 tex. The plates noted by “A1” and “B1” had four layers and the stacking sequence was [0/90]_4_, whereas the plates noted by “A2” and “B2” had eight layers and the stacking sequence was [0/90]_8_.

The plates A and B were manufactured with two different wf ratio of reinforced material (see Table 1 for A1, A2, B1 and B2 sample properties). The matrix used was an epoxy-resin-type EPIKOTE MGS LR135/LH 135 from Hexion Company, Esslingen am Neckar, Germany. The resin and the hardener were mixed at wf of 100:35.

The plates A1 and B1 were manufactured using a special pressing device that is presented in Figure 2. To impregnate the GF fabric, wet technology was used. In both cases of the manufacturing procedure, a metal-plate-like mold was used. The active surface of the mold was treated by release chemical agents: Mold Sealer type S31 from Jost Chemicals Company, Kościan, Poland (2 layers) and mold-release-type Frekote 770NC from Loctite Company, Düsseldorf, Germany (5 layers). After each layer deposition, we waited 10–15 min to ensure the liquid release substances were dry. In the next step, GFRP was applied on the mold surface layer by layer. After impregnation, the wet GFRP was covered with foil. This technology consists of pressing the wet composite by a vacuum system without the use of a vacuum pump. The mold with GFRP and foil was pressed by a cylinder to eliminate the excess resin from the GFRP. The cylinders are rotating, and the plates pass through the space between the cylinders. The cylinders are rotating in opposite directions as indicated in the Figure 2. The plate moves through the gap between the cylinders, which can be adjusted in height (depending on the thickness of the plate and the composite). GFRP is placed on top of the plate considered as the mold. The cylinders rotate by means of an electric motor with a reducer. The device works on the principle of a calendaring installation. The mold is handled manually, on the transport rollers up to the cylinders that take it due to the rotation movement. After passing through the cylinders, at the edges of the mold, the resin excess is removed. The innovative idea of this technology is to press the composite material onto the mold with an external force applied to the plastic foil covering the GFRP material. By pressing, the excess resin is removed towards the edges of the mold and the air bubbles from the GFRP material are removed, too. The excess resin seals the edges of the mold, so that there is no possibility for air to enter the composite material and change its structure. With the elimination of a quantity of resin, the volume of the composite material between the mold and the foil can be decreased. Negative pressure is formed between the mold and the plastic foil. This way, we use the atmospheric pressure to press the composite material without using a vacuum pump. In this way, the volume of material decreases, and the air cannot get inside. The viscosity of the resin does not allow the air to enter the composite in the edges area. In a short time, resin is transformed into a gel, and step by step, the polymerization stage occurs.

After passing through the device, the composites were kept at 22 °C, for 24 h to cure. A heat treatment was applied for 8 h at 80 °C after curing.

For the preparation of the A2 and B2 plates, a hand lay-up impregnation process and vacuum bag method were used (Figure 3a). All eight GF layers were applied and impregnated on the mold (Figure 3b) and covered with a perforated plastic foil and fabric breather (Figure 3c), then they were covered with a vacuum bag foil. The border of the bag was sealed by a vacuum sealing type (Figure 3c).

For pressing the composite layers, a vacuum pressure of −0.9 bar was applied on the bag during the curing time. The curing time was 4 h 80 °C.

The same heat treatment was applied for all A and B GFRP plates. An oven was used, and all the plates were kept at 120 °C for 8 h. The GFRP plates were then heated at a temperature from 22 to 120 °C at 2 °C/min. ramp rate for 50 min. Finally, the plates were cooled with the oven for the same amount of time after it was switched off.

### 2.2. Machining and Measurements Methods

To prepare GFRP plates for drilling they were cut on WaterJet Combo abrasive water jet cutter into samples of specified dimensions 35 mm × 250 mm (6 samples were used from each type of GFRP plate). After that, 10 holes (corresponding to 5 repetitions × 2 technological parameters) were drilled in each sample a spacing of of 25 mm between the axis of the holes (Figure 4) on vertical machining center Avia VMC800HS (Avia, Warsaw, Poland) (Figure 5a) without coolant. A 2-edge carbide diamond coated drill produced by Seco (Erkrath, Germany) (SD205A-12.726-56-14R1-C2) was used for drilling (Figure 5b). The main dimensions of the drill used in research: diameter of 12.726 mm, a point angle of 120° and a total length of 124 mm. The holes in the samples were drilled with different technological parameters: a cutting speed v_c_ = 91, 182 and 273 m/min and a feed per tooth, f_z_ = 0.04, 0.08, 0.12 and 0.16 mm/tooth (according to Table 2). The drilling parameters were determined during preliminary tests of GFRP drilling. As the main purpose of the research was to measure the cutting-force components, a special stand test was used, i.e., a 9257B dynamometer from Kistler (Winterthur, Switzerland) (Figure 5c). The main element of the stand was a piezoelectric dynamometer used to measure the cutting-force components F_x_, F_y_, F_z_, as well as a signal conditioning system, a DAQ module with an integrated A/D card and dedicated software (Figure 6). Delamination was determined by the delamination factor. In order to determine it, the diameters of the drilled holes and the delamination diameters were measured using the Keyence VHX-5000 optical microscope (Keyence, Itasca, MN, USA).

The delamination was quantified using the delamination factor and was defined as [31,32]:(1)$$F_{d}=\frac{D_{max}}{D_{nom}},\;[–]$$
where: $F_{d}$ is the delamination factor and $D_{max}$ is the maximum delaminated diameter drawn from the centerline of the hole nominal diameter $D_{nom}$ (according to Figure 7).

The diameters were measured using the software provided with the Keyence VHX-5000 optical microscope at 20× magnification.

### 2.3. Artificial Neural Network

Based on the obtained experimental results of experimental tests, models of the amplitude of the cutting-force component F_z_ (AFz) were developed. Only the component of the cutting force F_z_ was an object of prediction models, because it plays the largest role in the drilling process and assumes the largest values. The prediction models were designed using artificial neural networks in the Statistica 13 software. Variable technological parameters, i.e., a cutting speed v_c_ [mm/min] and a feed per tooth f_z_ [mm/tooth] and the type of the analyzed material (M) were adopted as input neurons, and the amplitude of the cutting-force component F_z_ (AFz) was taken as the output neuron. Figure 8 shows the schematic structure of the modeled neuron networks.

A shallow neural network with one hidden layer was used for modeling. The Multilayer Perceptron (MLP) and Radial Basis Function (RBF) neural network were used in the study. Different neuron activation functions were used for MLP: linear, exponential, logistic, tanh and sinus, together with different learning algorithms such as the BFGS gradient (Broyden–Fletcher–Goldfarb–Shanno), steepest descent training algorithm and conjugate gradient. The learning algorithm for the RBF was a RBFT, with a Gaussian hidden neuron activation function and a linear function for the output layer. The number of neurons in the hidden layer (2–10) was selected experimentally. The dataset was split in a proportion of 75% and 25% (for training and validation data, respectively). The test set was not included due to the small amount of data [33,34]. Network quality was assessed based on the value of the correlation coefficient R that was calculated in accordance with the formula:(2)$$R\left(y^{\prime },\;y^{*}\right)=\frac{cov\left(y^{\prime },\;y^{*}\right)}{\sigma_{y^{\prime }}\sigma_{y^{*}}}\;\;\;\;\;\;\;\;\;\;\;\;\;\;R\epsilon <0,1>$$
where: $\sigma_{y^{\prime }}\;$ is the standard deviation of the experimental values, $\sigma_{y^{*}}$ is the standard deviation of the predicted values.

In addition to the regression coefficient R, the mean square error (MSE) was also taken into account when selecting the best network, and its value was calculated according to the formula:(3)$$MSE=\frac{1}{n}\overset{n}{\underset{n=1}{\sum }}{\left({\hat{y}}_{i}-y_{i}\right)}^{2}\;$$
where: $y_{i}$ is the real value of the amplitude of the cutting-force component F_z_ for the i-th observation, and ${\hat{y}}_{i}$ is the value of the amplitude of the cutting-force component F_z_ for the *i*-th observation obtained as a result of the prediction. Similar analyses and neural networks were used in the drilling process case for other materials [35].

## 3. Results Discussion

### 3.1. Cutting Forces

Figure 9 illustrates the typical characteristics of the course of changes in the value of the cutting-force component F_z_ as a function of the time t in drilling a GFRP sample. This is an example diagram showing obtained values of the F_z_ component (also known as a feed force or thrust force), which plays a major role in the drilling process. Three zones can be identified in the force trajectory: entry zone, main drilling zone and exit zone. In the entry zone, the tool enters the workpiece, and an almost linear increase in cutting-force component F_z_ is observed. In the main drilling zone, the value of the cutting-force component F_z_ is more stable due to full engagement of the cutting edges [10]. In the exit zone, where the drill exits the workpiece, a decrease in the values of the cutting-force component F_z_ is observed. Periodic peaks that are observed in the graph occur when the composite fibers undergo cutting.

A fragment of the graph in Figure 10 was enlarged to mark the Simple Moving Average (SMA). The SMA was used to smooth the course of the F_z_ component of the cutting force.

Below, we present the diagrams illustrating a relationship between the amplitude value (which was determined as the difference between the maximum value and the minimum value of the force signal) of individual cutting-force components F_x_, F_y_, F_z_ and certain technological cutting parameters, i.e., the cutting speed v_c_ (Figure 11, Figure 12 and Figure 13) and the feed per tooth f_z_ (Figure 14), in the drilling process of the GFRP composite material. The monitoring and control of the forces can improve the drilling process. In particular, when safety-critical components are involved, process analysis based on measurements of process forces is the key factor that ensures maximum production reliability. The standard deviation is marked in all diagrams as a measure of scattering of the results.

Figure 11 shows the effect of drilling conducted with different values of the cutting speed v_c_ and with a constant feed per tooth f_z_ on the amplitudes of the cutting-force component F_x_ for four different types of GFRP materials. Noticeable differences in the amplitude values of the F_x_ cutting-force component occur at the cutting speed v_c_ = 91 and 182 m/min, while for the cutting speed v_c_ = 273 m/min, regardless of the GFRP type, the force-amplitude values are within a similar range. The highest amplitudes of the cutting-force component F_x_ achieved for material B1 and were: 298 N for v_c_ = 91 m/min, 234 N for v_c_ = 182 m/min and 177 for v_c_ = 273 m/min, respectively. For this material, the decrease in the amplitude of the cutting-force component F_x_ occurred proportionally with an increase in subsequent cutting speeds. The lowest amplitude of the cutting-force component F_x_ was obtained for materials A1 (130 N at v_c_ = 182 m/min) and A2 (134 N at v_c_ = 91 m/min) and B2 (137 N at v_c_ = 91 m/min). For material A2, the amplitude of the cutting-force component F_x_ increased with increases in the cutting speed (contrary to the B1 material). It also seems important to compare the A1 and B1 materials that were produced by the same technology using the special pressing device. These materials have a similar thickness and the same number of layers but differ with the fiber type. With lower cutting speeds (91 and 182 m/min), the amplitude of the cutting-force component F_x_ increases for plain woven fibers (material B) compared to twill woven fibers (material A), by about 63% and 80%, respectively, and it stabilizes at higher cutting-speed values.

A relationship between the cutting speed v_c_ and the cutting-force component F_y_, in a drilling process conducted with a constant feed per tooth of f_z_ = 0.16 mm/tooth for different GFRP materials, is presented in Figure 12. Due to the fact that the share of the cutting-force component F_y_ in the drilling process is lower than those observed for other components, its values for the different tested materials and technological parameters are lower, too. The maximum values of the amplitude of the cutting-force component F_y_ were obtained (similarly to the F_x_ component) for material B1 and were equal to 248 N (for v_c_ = 182 m/min) and 223 N (for v_c_ = 91 m/min). The minimum value of F_y_ was obtained for the B2 material in the drilling process conducted with the lowest cutting speed, and it was equal to 124 N. Similar to the F_x_ component, the amplitude of the cutting-force component F_y_ increased with increases in the cutting speed for material A2.

Due to the fact that the share of the cutting-force component F_z_ is the largest in the drilling process, the values of its amplitude are also the highest. A relationship between the cutting speed v_c_ and the amplitude of the cutting-force component F_z_ for the GFRP materials drilled with a constant feed per tooth of f_z_ = 0.16 mm/tooth is presented in Figure 13. The maximum values of the amplitude of the cutting-force component F_z_ were obtained for materials B1 and B2 and were 300 N and 303 N, respectively. The minimum value of 185 N was obtained for A1 material in a drilling process conducted with the lowest tested cutting speed. A comparison of materials A1 and B1 that were produced by the same technology showed that they had a similar thickness, the same number of layers but were made of different fiber types demonstrates, that in the drilling process conducted with lower cutting speeds (91 and 182 m/min), the amplitude of the cutting-force component F_z_ increased by about 26–38% when plain woven fibers were used (material B) compared to twill woven fibers (material A). For the highest cutting speed, the amplitude of the cutting-force component F_z_ becomes stable and is comparable for materials A1, A2 and B1. It is worth drawing attention to material B2, where the amplitudes of the cutting-force component F_z_ are the highest and the most comparable, regardless of the technological parameters defined by cutting speed.

The factor influencing the value of the cutting-force component F_z_ is also the sample thickness, because thicker samples will be stiffer compared to thinner ones. In order to assess the effect of the sample thickness on the cutting-force component F_z_, it would be necessary to perform tests on a larger number of samples with different thicknesses. In the presented studies, samples were positioned in such a way that they could flex (there was no central support; the plates were fixed only on the edges like in industrial drilling for assembly of the parts with rivets).

Figure 14 shows the influence of feed per tooth f_z_ on the amplitude value of cutting-force component F_z_ for different types of GFRP. It can be observed that the amplitude of the cutting-force component F_z_ increases with increases in the feed per tooth f_z_ for each tested material. Similar observations were made in [36] with respect to the drilling of unidirectional CFRP. Additionally, T. Panneerselvam [23] concluded that the parameter that has the greatest impact on delamination is the feed rate. The lowest amplitude values of the cutting-force component F_z_ were obtained for the lowest tested feed per tooth value of 0.04 mm/tooth for all tested materials (A1 = 132 N, A2 = 154 N, B1 = 121 N, B2 = 218 N), and it can be observed that the materials produced with the use of a special pressing device are characterized by lower values of the cutting-force component F_z_. This agrees with the results reported in [35,37], where it was also feed that had the greatest impact on the amplitude values of the cutting-force component F_z_. A comparison of the lowest and the highest rates of feed per tooth reveals that the largest increase in the amplitude of the cutting-force component F_z_ (by about 147%) was obtained for material B1, while the smallest for B2 (the amplitude value increased by about 38% increase).

### 3.2. Simulation

As a result of modeling based on network errors and the quality of training and validation, one network of each type (RBG and MLP) was selected for the amplitude of the cutting-force component F_z_ (denoted by AFz in the diagrams). Obtained modeling results and networks parameters are presented in Table 3.

One hundred networks of each type were assessed during modeling. The most suitable parameters for the MLP network were found for a network with two neurons in the hidden layer, obtained in 1320 iterations and with a logistic activation function of the hidden and linear layers for the output layer, whereas the most suitable parameters for the RBF network were found for a network with nine neurons in the hidden layer. The quality of training and validation for the MLP and RBF networks exceeds 0.95. In Table 2, we have provided the values of the correlation coefficients R (for the entire dataset) for experimental modeling results. The results show that correlation between the experimental data and the data provided for the MLP network is very high (above 0.98).

For a more detailed comparison of the RBF and MLP networks, Figure 15 shows the correlation diagram of the amplitudes of the cutting-force component F_z_ that were obtained experimentally and via modeling with the RBF and MPL networks.

Based on an analysis of the graph presented above, as well as network errors and the quality of training and validation, it can be concluded that higher-fit results of the amplitude of the cutting-force component F_z_ were obtained for the MLP network. As a result of modeling, it was possible to predict the amplitude of the cutting-force component F_z_, and for this purpose, a trained MLP network was used. By introducing new data into the Statistica program (input data: v_c_ [m/min], f_z_ [mm/tooth] and material type), the generated network returned the predicted amplitude values of the cutting-force component F_z_. Obtained results of the network performance are presented in Figure 16: for material A1 in Figure 16a, for material A2—Figure 16b, for material B1—Figure 16c, and for material B2—Figure 16d.

The modeling results of the amplitude of cutting-force component F_z_ [N] and the prediction made demonstrate that the obtained network is characterized by a satisfactory prediction ability of this parameter, because the correlation coefficient is above 0.98 and the size of errors does not exceed 15%.

### 3.3. Delamination of GFRP Materials

The employed measuring for the determination of delamination factor is shown in Figure 17.$\;D_{max}$ was drawn from the centerline of the $D_{nom}$ hole and was measured with the Keyence VHX-5000 microscope software. After that, the delamination factor was calculated in accordance with Equation (1).

Figure 18 shows the peel-up delamination factor values in the drilling process conducted with different feeds per tooth f_z_ and constant cutting speed v_c_ for different types of GFRP. Regardless of the tested material, for the lowest feed value, the delamination factor values are the lowest and most comparable (A1 = 1.05, A2 = 1.05, B1 = 1.04, B2 = 1.08). The diagram also reveals that feed per tooth has a significant impact on the delamination factor, as the delamination factor increases with an increase in the feed per tooth for all materials, which agrees with the results reported in [27]. A comparison of the lowest and the highest values of feed per tooth demonstrates that the smallest increase in the delamination factor amounting to about 12,5% is obtained for B1 material, while the highest delamination factor increase is obtained for B2 material and amounts to about 24%.

A comparison of the results presented in Figure 14 and Figure 18, where the relation between the feed per tooth f_z_ and the component of the cutting force F_z_ and delamination factor is presented, demonstrates a certain relation for all GFRP types. With the increase in the feed per tooth f_z_, both the delamination factor and the value of cutting-force component F_z_ increase. An increase in delamination and cutting force is caused by the cutting edge. The drill edge is in direct contact with the tested sample layer, which is extruded the fastest in comparison to the cutting. The cutting force causes greater thrust on the tested sample, which leads to the higher bending of the sample and can lead to the crack initiation and delamination development.

The relationship between the cutting speed and the delamination factor differs from that established for feed per tooth, as shown in Figure 19. No dependence can be observed between delamination factor increase or decrease in relation to cutting speed. For the material A2, the differences in the delamination factor values for different cutting speeds amount to a value of 0.01. A similar observation can be made for B1. The highest delamination factor value of 1.61 was obtained for B2 material in the drilling process conducted with v_c_ = 91 m/min. The delamination factor value decreased with increases in the cutting speed. The lowest delamination factor values were obtained for A1 = 1.13 (at v_c_ = 91 m/min) and for B1 = 1.16 (at v_c_ = 91 m/min). This is unusual, as study [27] showed that delamination depended on a higher cutting speed. For the material A1, the delamination factor increases with a cutting speed increase of 8% (at v_c_ = 182 m/min) and 3% (at v_c_ = 273 m/min), respectively.

Figure 20 shows the images of holes drilled in different GFRPs with a constant feed per tooth of f_z_ = 0.016 mm/tooth and three different cutting speeds v_c_ (the images are in a scale of 2:1). It can be observed that for the same drilling technological parameters of different materials, the influence of fiber type in the composite and the production method on delamination and defects in the form of torn fibers on the machined edges are noticeable. The smallest damage to the edges of the machined holes can be observed for materials A1 and B1, while the largest is observed for B2; these materials were made of the same fiber but differ in the number of layers and the manufacturing method. A similar observation can be made for materials A1 and A2, both of which were made of the same fiber, but had a different number of layers and different manufacturing methods—less delamination is observed for material A1, for which a special pressing device was used.

## 4. Conclusions

The novelty of this study consists of the preparation of GFRP using a new technique and the study of the machinability of these materials. Since the machinability of these materials is yet to be investigated, our study contributes to the knowledge in the field. The optimal parameters of the drilling process, which leads to low delamination phenomena, were determined.

The results of the study investigating the cutting forces and delamination in drilling GFRPs leads to the following conclusions:The cutting-force component F_z_ has the greatest influence on the total cutting force compared to other components;The lowest values of cutting-force component F_z_ were obtained for material A1 and the highest for material B2 (an increase of approximately 64%);A comparison of materials A1 and B1 demonstrated that, especially for lower cutting speeds, lower values of amplitude of the cutting-force component F_z_ are obtained for materials consisting of twill woven fibers (material A). In this case, the GF fabric is finer (280 g/m^2^) and is made of thinner threads;The feed per tooth f_z_ has an impact on the cutting-force component F_z_. For all tested GFRP materials, an increase in feed per tooth led to an increase in the amplitude of the cutting-force component F_z_;The lowest amplitude values of the cutting-force component F_z_ were obtained for the lowest tested feed per tooth value equal to 0.04 mm/tooth for all tested materials. It can be observed that the materials produced with the use of a special pressing device are characterized by lower values of the cutting-force component F_z_;Feed per tooth also has the greatest impact on delamination. The delamination factor increases with an increase in feed per tooth f_z_ for all tested GFRP materials;The lowest delamination factor increase, of about 12.5%, was obtained for the B1 material;Based on the images of the drilled holes, it can be concluded that the smallest amount of damage to the edges of the drilled holes was observed for materials A1 and B1. This proves that the manufacturing of composites with the use of special pressing device has a positive effect on delamination;The minimal delamination was obtained for B1 that was machined with the following machining parameters: v_c_ = 182 m/min and f_z_ = 0.04 mm/tooth;With the increase in the feed per tooth f_z_, both the delamination factor and the value of F_z_ cutting-force component increase for all types of GFRP;Since the error margin of the obtained numerical results does not exceed 15%, it can be concluded that artificial neural networks are a suitable tool for modeling cutting-force amplitudes depending on the cutting speed v_c_ and the feed per tooth f_z_.

Apart from the technological parameters of drilling, other factors that affect the cutting forces and delamination in this process include the manufacturing method of GFRP composites and the type of material. The materials manufactured using a special pressing device were characterized by a lower F_z_ cutting-force component in the drilling process and higher quality of the machined holes.

## Figures and Tables

**Figure 1 materials-15-08597-f001:**
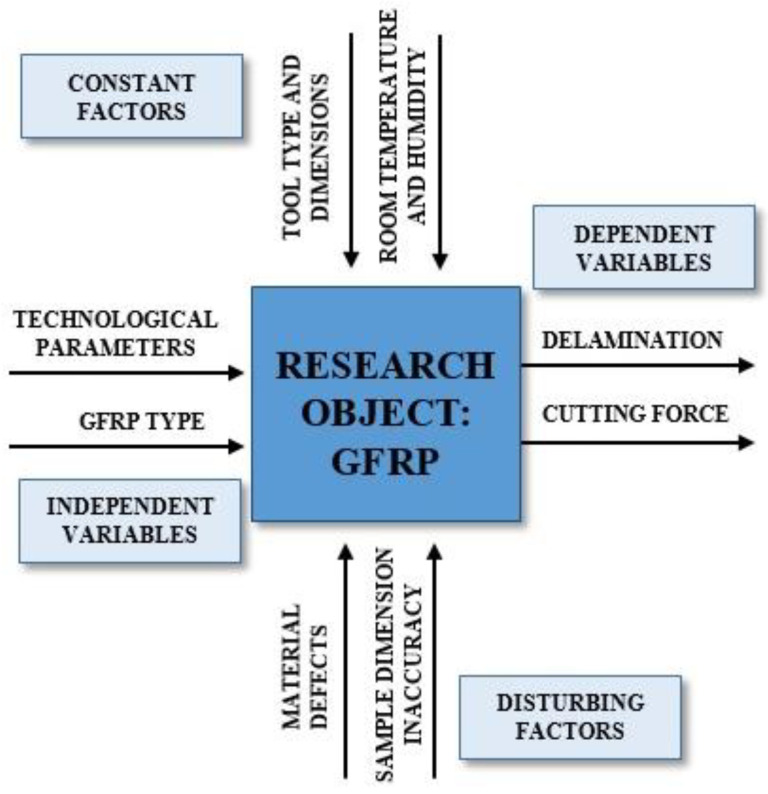
Research plan for the analysis of cutting forces and delamination during drilling of GFRP.

**Figure 2 materials-15-08597-f002:**
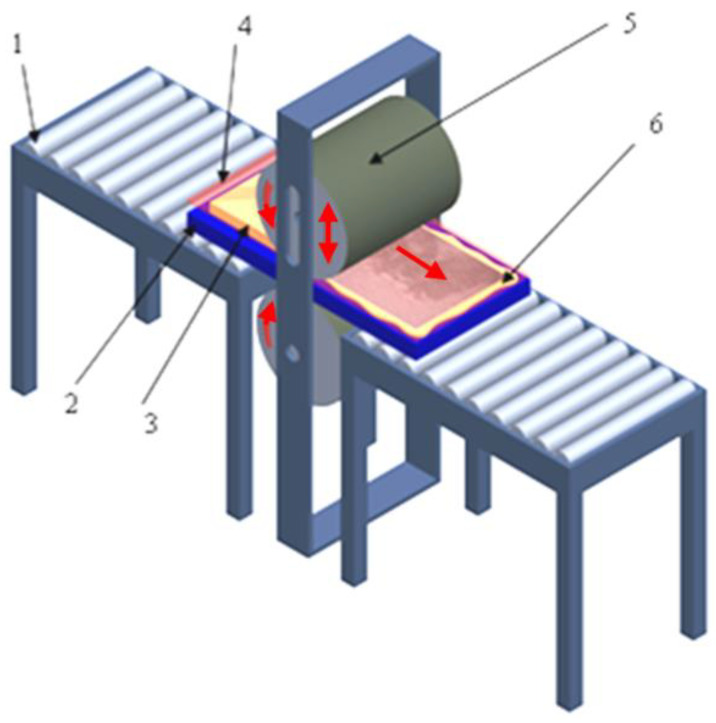
The scheme of GFRP pressing device: 1—worktable, 2—mold, 3—composite layers, 4—foil, 5—pressing cylinder device, 6—resin excess.

**Figure 3 materials-15-08597-f003:**
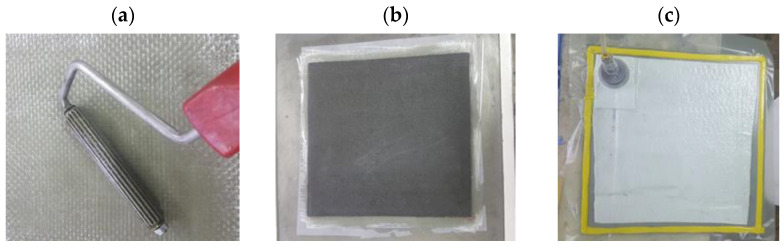
Vacuum bag technology applied for B2 GFRP plates.

**Figure 4 materials-15-08597-f004:**
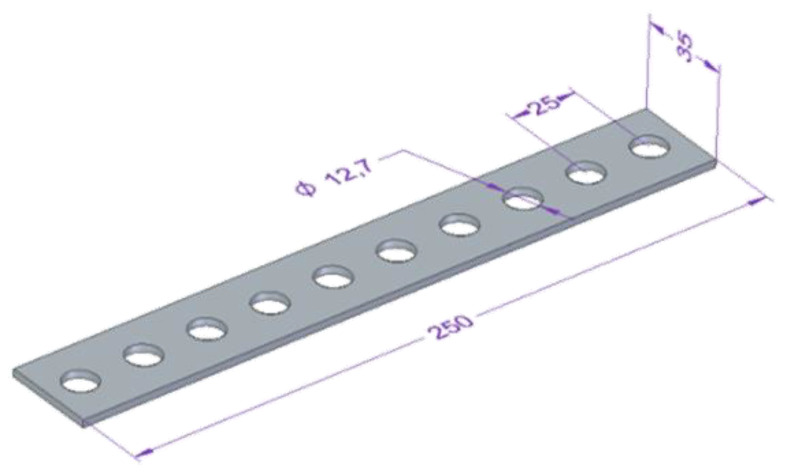
Dimensions of drilled samples.

**Figure 5 materials-15-08597-f005:**
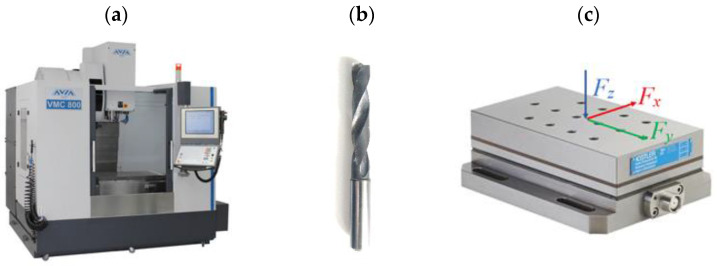
Equipment used for drilling: (**a**) vertical machining center Avia VMC800HS, (**b**) carbide-diamond-coated drill, (**c**) Kistler dynamometer.

**Figure 6 materials-15-08597-f006:**
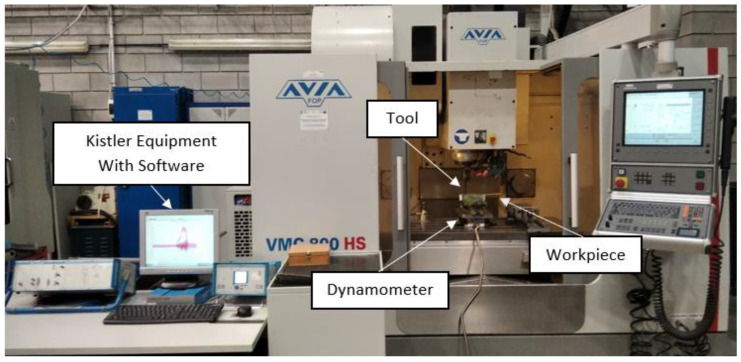
Test stand for measuring cutting forces.

**Figure 7 materials-15-08597-f007:**
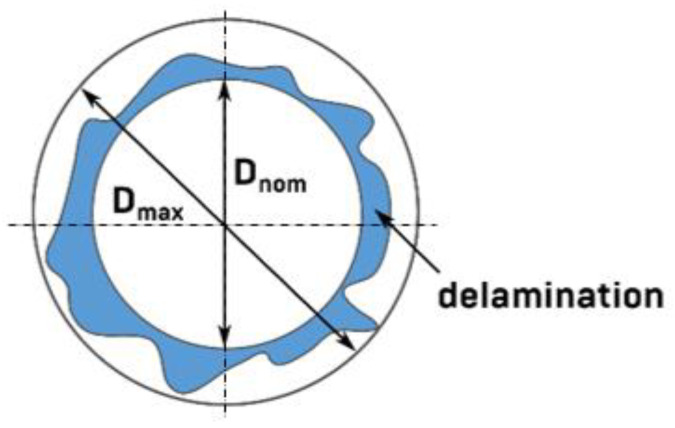
Delamination factor scheme.

**Figure 8 materials-15-08597-f008:**
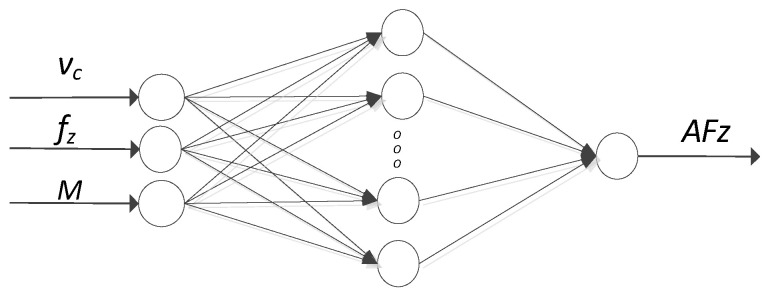
Schematic structure of the modeled neuron networks.

**Figure 9 materials-15-08597-f009:**
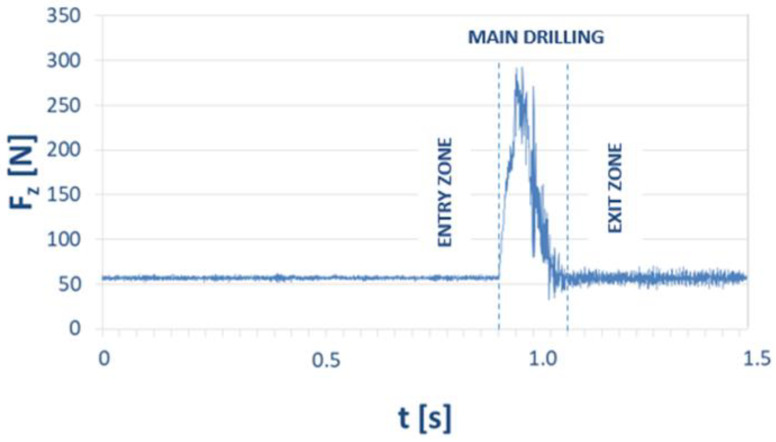
Example of time course of the cutting-force component F_z_ for B2 material for the following technological parameters: cutting speed v_c_ = 273 m/min, feed per tooth f_z_ = 0.16 mm/tooth.

**Figure 10 materials-15-08597-f010:**
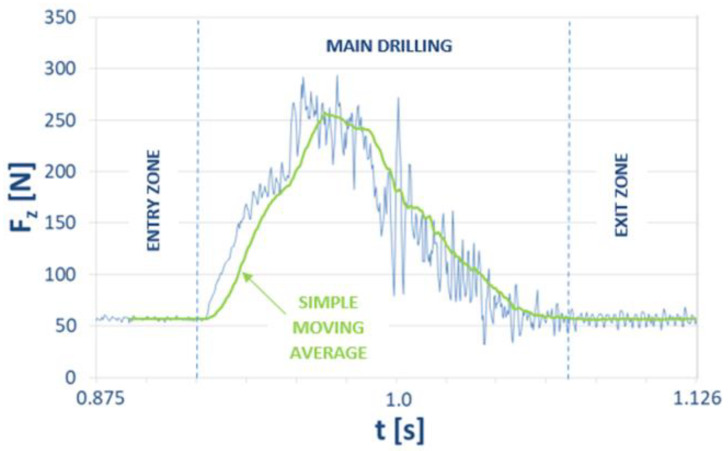
Enlarged fragment of the graph showing the time course of the cutting-force component F_z_ in Figure 9, with a marked course of simple moving average.

**Figure 11 materials-15-08597-f011:**
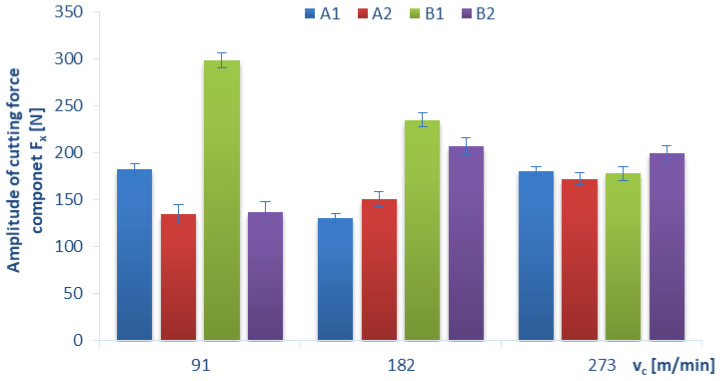
Relationship between the amplitude of the cutting-force component F_x_ and the cutting speed v_c_ in a drilling process conducted with a with constant feed per tooth of f_z_ = 0.16 mm/tooth, for different GFRP materials.

**Figure 12 materials-15-08597-f012:**
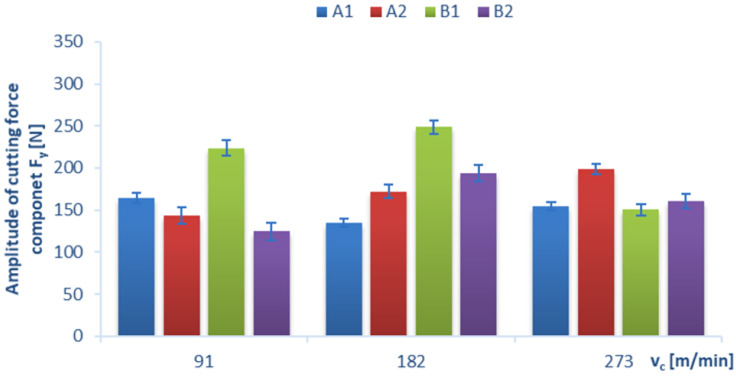
Relation between amplitude of cutting-force component F_y_ and cutting speed v_c_, with constant feed per tooth f_z_ = 0.16 mm/tooth for different GFRP materials.

**Figure 13 materials-15-08597-f013:**
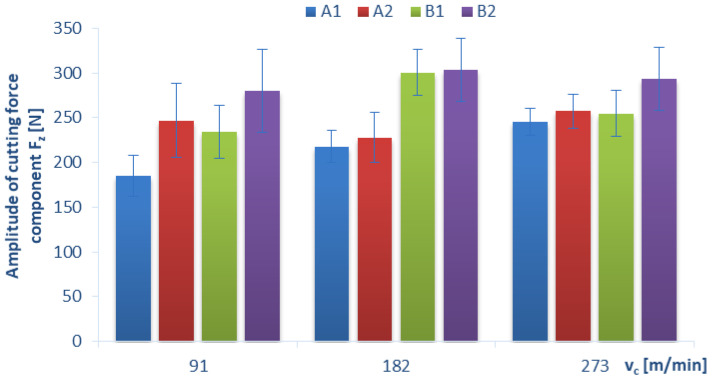
Relationship between the amplitude of cutting-force component F_z_ and the cutting speed v_c_ in a drilling process conducted with a with constant feed per tooth of f_z_ = 0.16 mm/tooth for different GFRP materials.

**Figure 14 materials-15-08597-f014:**
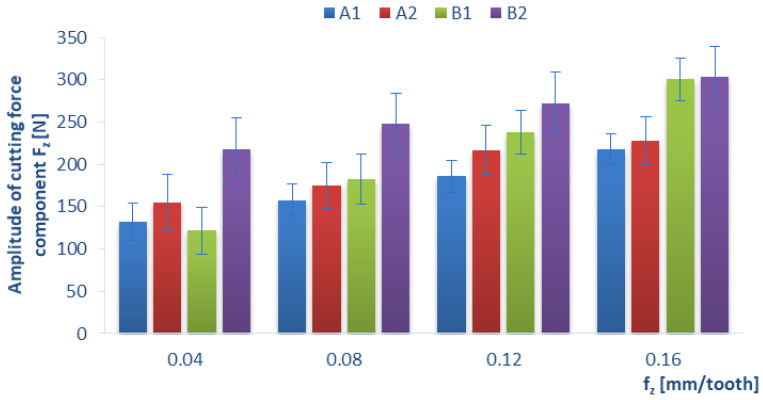
Relationship between the amplitude of the cutting-force component F_z_ and feed per tooth f_z_ in a drilling process conducted with a cutting speed of v_c_ = 182 m/min for different GFRP materials.

**Figure 15 materials-15-08597-f015:**
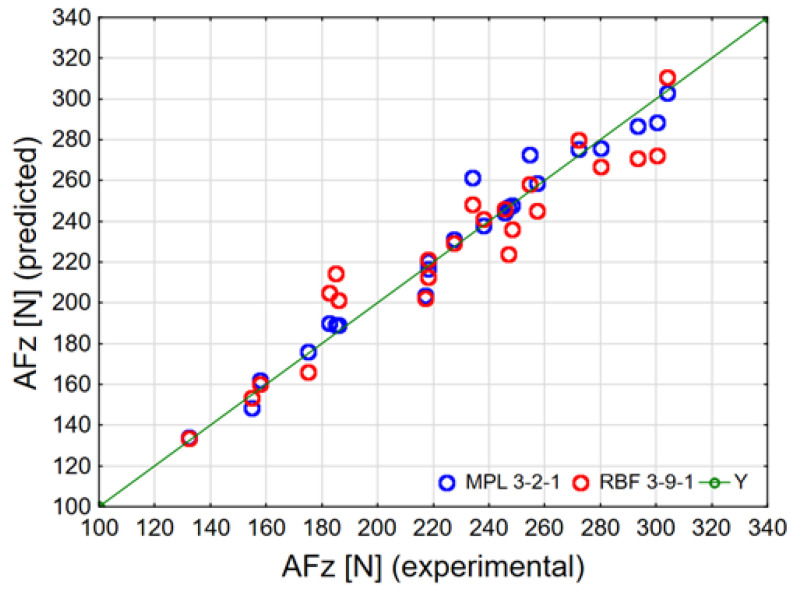
Correlation graph comparing the modeling and experimental results of the amplitude of cutting-force component F_z_ (AFz).

**Figure 16 materials-15-08597-f016:**
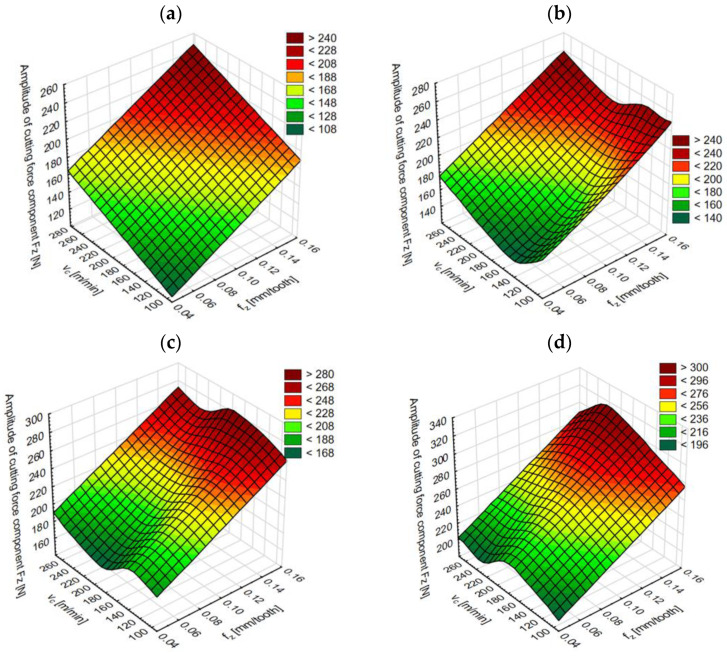
Network performance results of the MLP 3-2-1 for the materials: (**a**) A1, (**b**) A2, (**c**) B1, (**d**) B1.

**Figure 17 materials-15-08597-f017:**
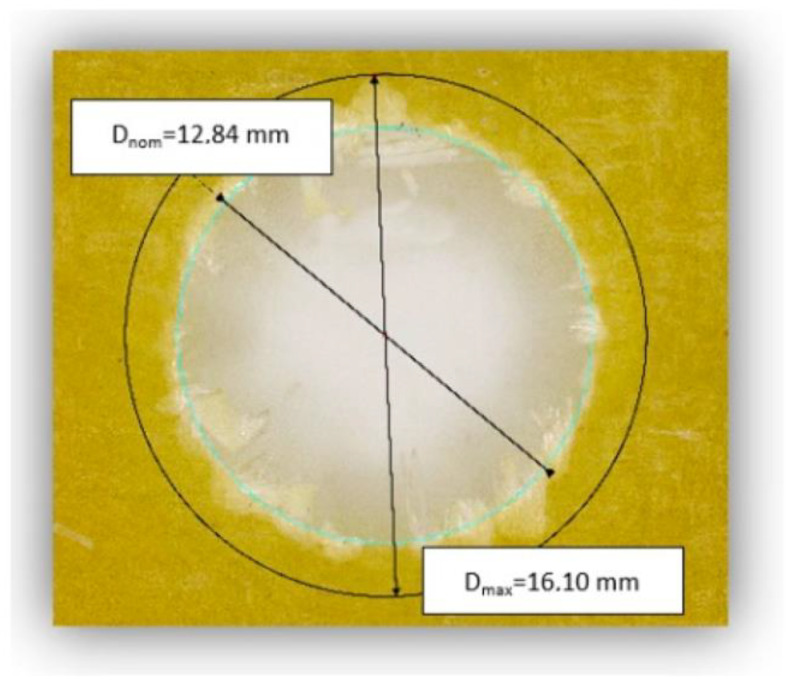
Measurement of D_nom_ and D_max_ for the determination of $F_{d}$.

**Figure 18 materials-15-08597-f018:**
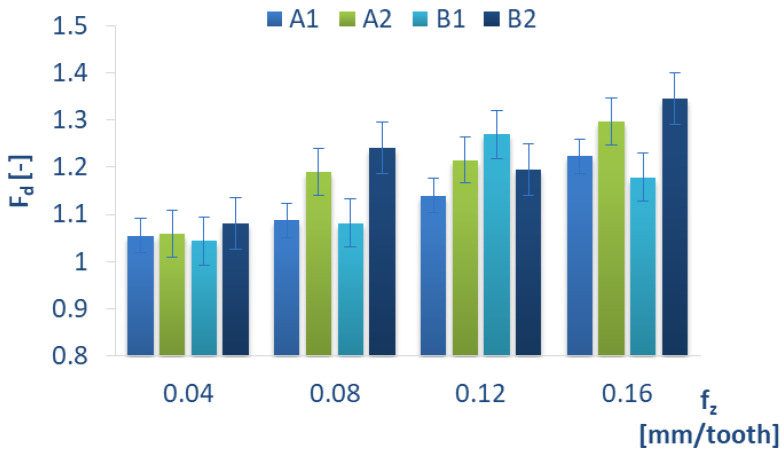
Delamination factor $F_{d}$ for different feed per tooth f_z_ and constant cutting speed v_c_ = 182 m/min for different GFRP materials.

**Figure 19 materials-15-08597-f019:**
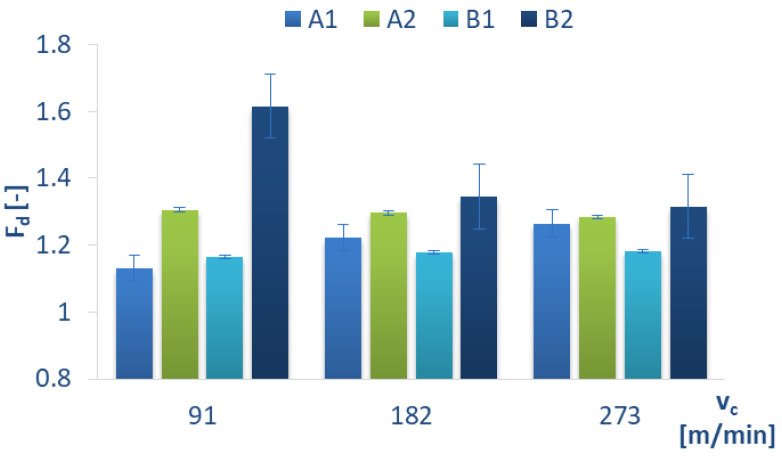
Delamination factor versus different cutting speeds v_c_ and constant feed per tooth f_z_ = 0.016 mm/tooth for different GFRP materials.

**Figure 20 materials-15-08597-f020:**
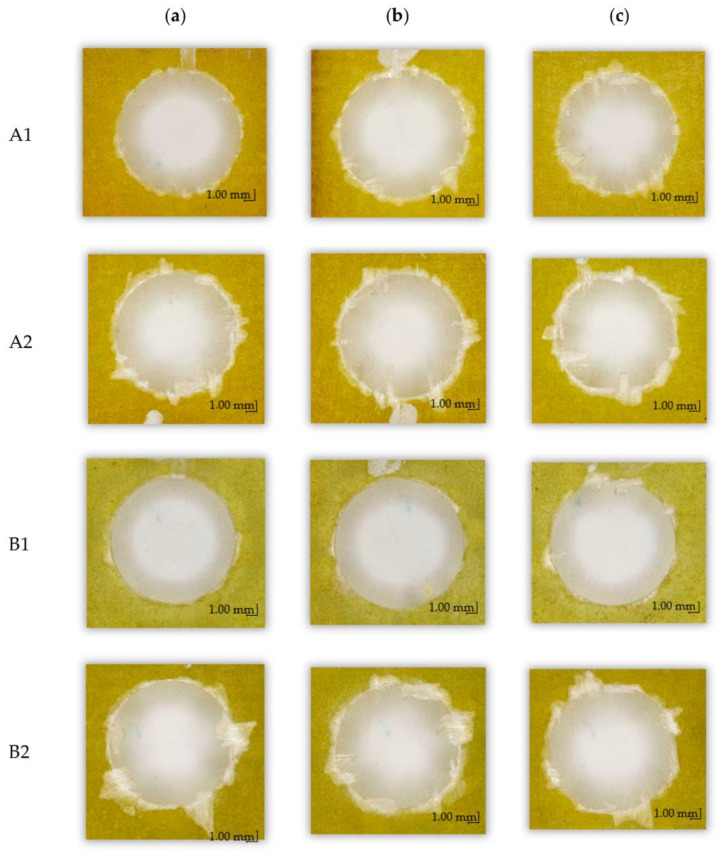
Peel-up delamination images of holes drilled in different GFRPs with a constant feed per tooth f_z_ = 0.016 mm/tooth and different cutting speeds v_c_: (**a**) 91 m/min, (**b**) 182 m/min, (**c**) 273 m/min.

**Table 1 materials-15-08597-t001:** GFRP plates properties.

Notation	A1	A2	B1	B2
Wf [%]	50	64	50	64
Thickness [mm]	1.5	2.1	1.55	2.1
Density [kg/m^3^]	1493	1729	1393	1767
Number of layers	4	8	4	8

**Table 2 materials-15-08597-t002:** Technological parameters of the GFRP composite drilling.

Cutting Speed v_c_ [m/min]	Feed Per Tooth f_z_ [mm/tooth]	Number of Holes	Number of Sample 35 × 200 [mm]
91	0.04	5	1
	0.08	5	1
	0.12	5	2
	0.16	5	2
182	0.04	5	3
	0.08	5	3
	0.12	5	4
	0.16	5	4
273	0.04	5	5
	0.08	5	5
	0.12	5	6
	0.16	5	6

**Table 3 materials-15-08597-t003:** Obtained modeling results and networks parameters.

Network Type	Network Name	Quality Training	Quality Validation	MSE Training	MSE Validation	Learning Algorithm	R_(i)_ Correlation
MLP	MLP 3-2-1	99.47%	95.92%	10.25	126.11	BFGS 1320	0.9842
RBF	RBF 3-9-1	95.48%	95.94%	85.65	149.37	RBFT	0.9553

## Data Availability

Not applicable.

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
