# Peer review of "Experimental Study and Artificial Neural Network Simulation of Cutting Forces and Delamination Analysis in GFRP Drilling"

_materials, 2022, doi:10.3390/ma15238597_

Round 1
Reviewer 1 Report
In this paper the authors presented a study regarding the drilling of GFRP plates. Considered very important research used for aviation industry and not only, the authors presented in detail the manufacturing of composite plates and machining procedure. The paper is well organized and contains important data regarding the drilling parameters for four different GFRP plates made with different types of fiber, weight fraction ratio, number of layers and manufacturing procedure, but with the same stacking sequence of the layers. The authors presented an innovative method to manufacture a composite plates, for what the authors obtained a patent. The main goal of the research was to define the drilling parameters for a specific composite material. I recommend some corrections in the text, as follows:
Rows 18-19: GFRP samples were made using two technologies: new method using a special pressing device and hand lay-up process. Please mention after Hand lay-up and vacuum bag technology.
Row 21: ,,drill produced by Seco.” Please mention SECO Company or what?
Row 40: ,,glass fiber (GF)”, Please use capital letters before the abbreviation Glass Fiber (GF) in all the text.
Row 43: low weight, you can use the low density.
Table 1. and 2. Glass fiber plates properties. Please use the GFRP plates properties instead Glass fiber. Please use different title for the Table 2. Please put some space after the Table 1. and Table 2.
Row 122: 3.1 Materials should be 2.1 Materials
Row 167: please write -0.9 bar, not -0.9 Bar.
Row 172: ,,At the end the plates was cooled...” Please correct to “At the end the plates were cooled...”
Row 174: 3.2 Machining method should be 2.2 Machining method.
Row 203: 3.3 Artificial Neural Network should be 2.3 Artificial Neural Network
Please center the figures 4, 6, 7 and all in the text.
3.3. Delamination. Please state Delamination of GFRP Materials.
Row 371: Fiber Reinforced Polymer (FRP). You already used
abbreviation GFRP in Abstract. Please use GFRP.
Row 436: ,, thickness of the sample” please correct to samples.
Why such values of drilling parameters were chosen in experiment and this type of tool was used?
Reviewer 2 Report
This work mainly studied the cutting force and delamination behavior in drilling of the glass fiber reinforced polymer (GFRP) composites, which was fabricated by a novel pressing device. The effects of cutting parameters (i.e., cutting speed and feed per tooth) on force and delamination were clarified. The cutting force component Fz was further analyzed and predicted based on artificial neural network simulation. Thus, the obtained results were beneficial to optimize the cutting parameters in drilling of GFRP composites. However, prior to further consideration by journal, the major revision is needed in order to meet the high requirement of journal, and hereinafter the main comments.
(1) In introduction, in this sentence, “The machinability of those materials was not has not been the subject of any research”, the English grammar should be corrected. In literature review, English tense and some descriptions should be double checked and improved.
(2) In introduction, the authors claimed that “The aim of research presented in this paper is to assess the influence of the various technological drilling parameters on the cutting forces occurring during machining and the delamination of GFRP composites. The novelty of that research is based on the used GFRP production methods based on special pressing device, which was patented”. The main research aim and the novelty in the current study are not the same, as described in this paragraph.
(3) In section 3.2, the cutting forces in each group of parameters were tested once?
(4) In Fig.10 and Fig.11, the B1 material has the largest cutting forces than other materials when vc=91, and 182 m/min, while it almost has the smallest forces when vc=273 m/min. Please give the further explanation. In addition, in Fig.12 and Fig.13, it was the B2 material that has the largest forces under three cutting conditions (i.e., different vc and fz values). Therefore, the material behavior during drilling is very confusing.
(5) According to Figs.11-13, the cutting forces during drilling of GFRP have three components. Why only Fz considered in simulation?
(6) Fig.19 gives the delamination images of GFRP, but the image quality is poor, and the defects cannot be identified using current images. Please provide some SEM and OM images with high quality if possible.
(7) The conclusions should be concised further.
Reviewer 3 Report
See the attachment for comments.

Reviewer 4 Report
The paper needs the following improvement:
1. The novelty of the current work is not clearly elaborated.
2. Due to the increasing interest in different industries, the scientific community is devoting remarkable efforts to these applications. Therefore, it is worth mentioning to refer the recent paper in the literature: https://doi.org/10.1007/s00170-019-04348-z
3. The authors should explain the reason for the selected parameters
4. Please add the motivation for why the authors were interested mainly in the thrust force and delamination. The reason for not including the other output parameters should be highlighted.
5. The motivation for the ANN should be included in the paper. As a guide, please refer to the recent paper: https://doi.org/10.3390/app10238633
6. Please include the Figure for the experimental setup, including the dynamometer
7. Please add a discussion section and reasons for the variation in thrust force and delamination due to the selected input parameters, i.e. cutting speeds and feed rates in this study
8. The authors should revise the conclusions accordingly.
Round 2
Reviewer 2 Report
In section 4, the conclusions should be simplified. Normally, this part presents the most important and novel findings in the investigation on 'Experimental study and artificial neural network simulation of cutting forces and delamination analysis of GFRP drilling'. However, the current conclusions are too long to get the key points.
Reviewer 3 Report
See the attachment for comments.

Reviewer 4 Report
Accepted.
